# Improved Detection of Adversarial Attacks via Penetration Distortion Maximization

## Abstract

This paper is concerned with the defense of deep models against adversarial attacks. We develop an adversarial detection method, which is inspired by the certificate defense approach, and captures the idea of separating class clusters in the embedding space to increase the margin. The resulting defense is intuitive, effective, scalable, and can be integrated into any given neural classification model. Our method demonstrates state-of-the-art (detection) performance under all threat models.

## 1 Introduction

Defending machine learning models from adversarial attacks has become an increasingly pressing issue as deep neural networks become associated with more critical aspects of society. Adversarial attacks can effectively fool deep models and force them to misclassify, using a slight but maliciously-designed distortion that is typically invisible to the human eye (Carlini & Wagner, 2017c; Athalye et al., 2018; Szegedy et al., 2013; Goodfellow et al., 2014; Kurakin et al., 2016). Despite numerous developments, defense mechanisms are still wanting.

Many interesting ideas have been proposed to construct defense mechanisms for adversarial examples. Among these are adversarial training (Zuo et al., 2020; Yan et al., 2018; Tramèr et al., 2017; Madry et al., 2017), ensemble methods (Strauss et al., 2017), and randomization (Dhillon et al., 2018) to name a few. These works consider both detection and resiliency. However, many of these defense ideas were found to be inadequate (Athalye et al., 2018; Carlini et al., 2019; Carlini & Wagner, 2017b; He et al., 2017). For example, adversarial training critically depends on the specific choice of adversarial attacks used to generate the adversarial training instances. As a result, often this method cannot withstand attacks based on different strategies. (Engstrom et al., 2018).

A more formal approach to adversarial defense is the *certification approach* (Hein & Andriushchenko, 2017), which is designed to provide a lower bound for the penetration distortion required to fool a given network. Certified defense methods are referred to as being either "exact" or "conservative". In exact methods no distortion smaller than the certification bound can penetrate the deep neural network (DNN) (Hein & Andriushchenko, 2017; Wong & Kolter, 2017; Wong et al., 2018; Cohen et al., 2019). In "conservative" methods, the bound is merely a relative metric for comparing DNN robustness to adversarial examples (Tsuzuku et al., 2018; Zhang et al., 2019; Ding et al., 2018). Both exact and conservative methods have been criticized for being computationally expensive and unscalable (Tjeng et al., 2018; Cohen et al., 2019).

It is interesting to view adversarial attacks through activation geometry in embedding layers. A trained deep classification model tends to organize instances into clusters in the embedding space, according to class labels. Classes with clusters in close proximity to one another, provide excellent opportunities for attackers to fool the model. This geometry explains the tendency of untargeted attacks to alter the label of a given image to that of an adjacent class in the embedding space as demonstrated in Figure 1a. Thus, if we can modify the model to increase the margin between clusters, while lowering (or not increasing) the activation sensitivity in the embedding space to input changes, we can make the network more immune to attacks. This embedding sensitivity can be quantified through a Lipschitz constant or directly via the Jacobian.

Inspired by certificate defense methods, we developed an adversarial detection method, that captures the above separation in embedding space intuition. Ideally, we would like to lower bound the distor-

tion, $\epsilon$, required by the adversary to force a DNN $F$ to misclassify $x + \epsilon$, where $x$ is an input image. We propose an approximation to such a bound which, while not formal, motivates a useful strategy for creating defense methods. The bound, $\epsilon \gtrsim \eta / ||J_F(x)||$, which is similar to other known bounds, is given in terms of $\eta$, where $\eta$ quantifies the "embedding margin" of the network, and $J_F(x)$, the Jacobian of $F$ with respect to $x$ (see details in Section 2). The embedding margin, for a given intermediate layer, is the minimal distance (under any $p$-norm) between two instances belonging to two different classes. This approximate relation motivates a strategy of *penetration distortion maximization* (PDM) whereby, we implicitly or explicitly maximize this lower bound without attempting to calculate it.

The notion of increasing a DNN classification margin has been discussed in several contexts Liu et al. (2016); Hoffer & Ailon (2015); Wen et al. (2016). To apply the PDM approach we propose two procedures to increase the embedding margin. These two methods are complementary in the sense that we can benefit by applying them together. In conjunction, we use the reverse cross-entropy method of Pang et al. (2018), which tends to smooth the Jacobian. Our adversarial detection mechanism is constructed by training a resilient classifier using the above three procedures; we then apply standard kernel density estimation (KDE) on the embedding layer (Feinman et al., 2017). We present an extensive empirical study focusing on the detection of adversarial examples under all threat models, in which we consider the FGSM, BIM, C&W and JSMA attacks. Our experimental procedure strictly adheres to the comprehensive evaluation desiderata proposed by Carlini et al. (2019). The results we obtain indicate that the proposed defense achieves state-of-the-art detection.

## 2 PENETRATION DISTORTION MAXIMIZATION

In this section, we explain the PDM strategy. Let $F$ be a neural classifier and let $x \in \mathbb{R}^{h \times w}$ be an image assumed to have class label $c = c(x)$. Let $\epsilon \in \mathbb{R}^{h \times w}$ be a vector representing an adversarial distortion for image $x$ such that the (successful) adversarial instance is $x_{adv} \triangleq x + \epsilon$ whose label is different from $c$; namely, $c_{adv} \triangleq F(x_{adv}) \neq c$. The attacker's goal is to find the smallest perturbation $\epsilon$ such that $F$ misclassifies $x$,

$$\min_\epsilon ||\epsilon||$$

$$\text{s.t. } F(x + \epsilon) \neq c(x) .$$

For a successful adversarial attack whose distortion is required to be small, in the spirit of (Ding et al., 2018; Tsuzuku et al., 2018; Hein & Andriushchenko, 2017; Zhang et al., 2019), we approximate a prediction for $x_{adv}$ using the first-order Taylor approximation

$$F(x_{adv}) = F(x + \epsilon) \overset{|\epsilon| \ll 1}{\approx} F(x) + J_F(x)\epsilon, \tag{1}$$

for vector-valued functions with $J_F(x)$ being the Jacobian of $F$. The same approximation applies to the output of any intermediate layer $\ell$. Denoting by $F_\ell(x)$ the output of layer $\ell$ we thus have,

$$F_\ell(x_{adv}) \approx F_\ell(x) + J_\ell(x)\epsilon.$$

For layer $\ell$, we define its *embedding margin*,

$$\eta_\ell \triangleq \underset{x_1, x_2, c(x_1) \neq c(x_2)}{\arg\min} ||F_\ell(x_1) - F_\ell(x_2)||.$$

Thus,

$$||J_\ell(x)\epsilon|| \approx ||F_\ell(x) - F_\ell(x_{adv})|| \geq \eta_\ell \tag{2}$$

The Frobenius norm used here is sub-multiplicative (proof can be found in Appendix A); namely,

$$||J_\ell(x)||||\epsilon|| \geq ||J_\ell(x)\epsilon||. \tag{3}$$

Combining (2) and (3) (and ignoring the approximation error) we lower bound the norm of the distortion $\epsilon$ in terms of the embedding margin and the norm of the Jacobian,

$$||\epsilon|| \gtrsim \frac{\eta_\ell}{||J_\ell(x)||}. \tag{4}$$

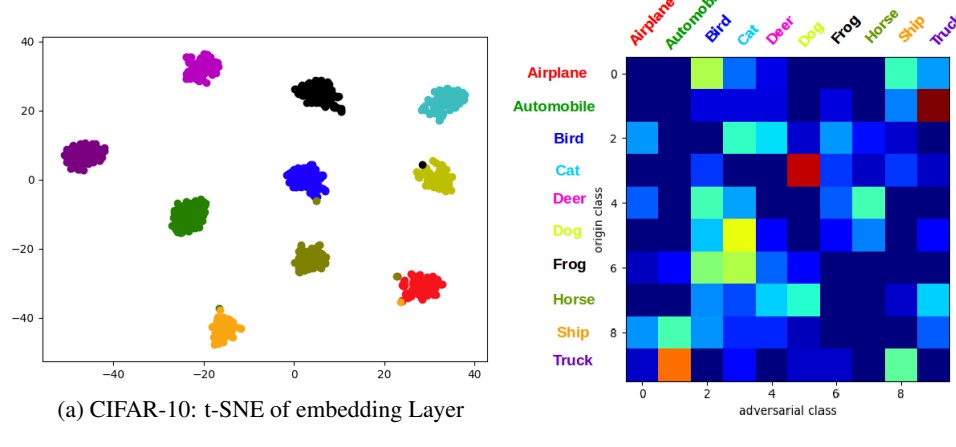

(a) CIFAR-10: t-SNE of embedding Layer

(b) Adversarial confusion histogram

Figure 1: Histogram of origin and target classes from C&W untargeted adversarial attack compared to embedding layer t-SNE

While the attacker's goal is to find a small distortion that "penetrates" another class, our goal as the defender is to create a resilient model that forces a larger distortion. The lower bound (4) motivates our *penetration distortion maximization* (PDM) method whereby the goal is to explicitly maximize the right side of (4) with respect to the embedding layer of the model $F$. To successfully apply this technique we must increase the embedding margin $\eta_l$ (while not increasing the norm of the Jacobian) and/or smooth the network to decrease the norm of the Jacobian $||J_\ell(x)||$. We note that similar and stronger, formal bounds, in terms of the Lipschitz constant, have been introduced by Tsuzuku et al. (2018), Zhang et al. (2019), Hein & Andriushchenko (2017) and Ding et al. (2018).

## 3 INCREASING RESILIENCY USING PDM

In this section we show how we use PDM, which is applied to the final layer that captures the full embedding of the network (often referred to as the "pre-logits"). We note that technically we can also apply PDM to any other layer in the model but defer such explorations to future work. The proposed approach consists of three components, which are described in this section. Two novel components are used to increase the margin, and the third is a known technique that is responsible for reducing the norm of the Jacobian.

Our approach for increasing the embedding margin relies on the observation that at higher embedding layers of a trained model, the embedding vectors (tensors) of instances tend to be structured in clusters according to class labels. This can be seen, for example, in Figure 1a where we observe the t-SNE visualization (Maaten & Hinton, 2008) of the embedding layer of a network trained for CIFAR-10. Moreover, we observe that an adversarial example created by an untargeted attack often obtains a class label whose cluster is in close proximity with the cluster of the original class. In Figure 1b we present a color matrix showing the adversarial label distribution obtained by the C&W attack (Carlini & Wagner, 2017a). For example, the color for the *Cat* and *Dog* entry is bright red indicating a frequent label change from cats (original) to dogs (adversarial), whose clusters are the closest. By increasing the margin between these clusters without increasing the norm of this layer's Jacobian ,we make it harder for an adversary to alter the label using distortion of the same magnitude.

Adopting ideas from cluster analysis, the increase in the embedding margin can be achieved by either increasing the distance between clusters or reducing the variance of each cluster. Let $\mu_c = \frac{1}{N_c} \sum_{i=1}^{N_c} z_i^c$ be the mean of each cluster, where $N_c$ is the number of samples from class $c$ and $z_i^c$ is

the embedding of sample $i$ from class $c$, and let $M$ be the number of classes. We thus have,

$$\text{Cluster Variance} \triangleq \sum_{c=1}^{M} \frac{1}{N_c} \sum_{i=1}^{N_c} ||z_i^c - \mu_c||_2$$

$$\text{Cluster Distance} \triangleq \frac{1}{M} \sum_{c=1}^{M} \frac{1}{M-1} \sum_{i \neq c}^{M} ||\mu_i - \mu_c||_2$$

To increase the margin, we would like to maximize the cluster distance and minimize the cluster variance, hence

$$\text{Margin Maximization Objective} = \text{Cluster Variance} - \text{Cluster Distance}.$$

A straightforward maximization of the cluster distance is problematic because the distance is potentially unbounded. However, we can proxy the distance using the angular distance between clusters. To this end, we use the cosine similarity. We now introduce two methods to optimize these components. We use a Siamese training procedure to maximize the cluster distance. The cluster variance is minimized by including a variance in the loss function.

## 3.1 SIAMESE TRAINING

To explicitly increase the embedding margin, we propose using Siamese training. We create a Siamese network (Bromley et al., 1994), where each sub-network is our classifier. The Siamese network has two input images denoted by $x_i^c, x_j^{\tilde{c}}$ and three outputs: two classification outputs and an auxiliary output for the cosine similarity between each sub-network's embedding. We introduce an additional loss term to force embeddings from different classes samples to have a cosine similarity of 0 or 1 otherwise. Formally,

$$\text{SiameseLoss} = \frac{z_i^c \cdot z_j^{\tilde{c}}}{||z_i^c||||z_j^{\tilde{c}}||} \stackrel{!}{=} \begin{cases} 1 & \text{if } c = \tilde{c} \\ 0 & \text{else} \end{cases}$$

## 3.2 REDUCE VARIANCE LOSS

Inspired by Szegedy et al. (2016), we include an additional loss term that penalizes large variance for each class' cluster individually. We refer to this component as the "reduce variance loss" (RVL). Formally,

$$\sigma_c \triangleq \frac{1}{N_c} \sum_{i=1}^{N_c} ||z_i^c - \mu_c||_2, \qquad \text{RVL} \triangleq \frac{1}{N_c} \sum_{c=1}^{N_{classes}} \sigma_c \tag{5}$$

The variance is estimated per class on each mini-batch, then averaged and minimized as part of the learning process.

## 3.3 REVERSE CROSS ENTROPY

We use the reverse cross-entropy loss introduced by Pang et al. (2018) to minimize the norm of the Jacobian. By labeling a sample with a "reverse" one-hot vector, we obtain

$$R_i^c = \begin{cases} 0, & \text{if } i = c \\ \frac{1}{(N_{\text{classes}} - 1)}, & \text{else}, \end{cases}$$

and using a reverse cross entropy loss

$$L_{\text{RCE}} = -R^c \log F(x).$$

Similar to label smoothing (Szegedy et al., 2016), this method smooths the classifier's gradients and prevents the network from becoming over-confident (Müller et al., 2019). Intuitively, the differentiation between two samples has a tighter upper bound, given the reverse labels $R^c$, than a one-hot labeling. We tested the gradient $L_2$ norm value on different layers. Comparing to the baseline model, the gradients were five to ten times smaller when using the RCE training process.

## 3.4 PDM TRAINING

A simultaneous application of the three components described above, which can robustify a classification model, is obtained by training the model using an appropriate loss function as well as a specialized mini-batch construction procedure. A pseudo-code of the training procedure including the loss function appears in Algorithm 1 under Appendix F. The code is self-explanatory for the most part. We note that an epoch begins by creating a Siamese counterpart for each image-label pair in a given batch. With probability $Q$, the Siamese sample is selected from the same class, and its cosine similarity label is set to 1. Otherwise (probability $1 - Q$), the Siamese sample is selected from a different class, and its cosine similarity label is set to 0. Notice that the Siamese and RVL (Equation 5) components of the loss function are computed from the embedding vectors of each mini-batch. The RCE component is calculated using the logits.

## 3.5 PDM VISUALIZATION

Using t-SNE to visualize the embedding space activation, Figure 2 illustrates the effect of each of the components of our defense method. Figure 2c demonstrates how well the RVL reduces the variance, while the Siamese training process made a more profound impact on the between-class distance as shown in Figure 2d.

While t-SNE is useful for visualization purposes, the aggressive dimensionality reduction may lead to misleading conclusions. To obtain quantitative evidence, we calculated the Davies–Bouldin index (DBI) (Davies & Bouldin, 1979), which scores clustering quality according to the distance between cluster centroids divided by the Euclidean distance between points within a cluster (lower score means better clustering). The DBI of the combined method is indeed the lowest at 0.23. (See the other DBIs in the figure.)

## 4 EXPERIMENTS

Following (Pang et al., 2018; Meng & Chen, 2017; Madry et al., 2017; Song et al., 2017; Dhillon et al., 2018; Samangouei et al., 2018) we evaluated our defense technique on the MNIST (LeCun et al., 1998) and CIFAR-10 (Krizhevsky & Hinton, 2009) datasets.

We adopted the detection method presented by Feinman et al. (2017), using a univariate Gaussian-based kernel density estimation (KDE), where the density was estimated using 1000 training sample embeddings per class. An input image is deemed adversarial if the distance to the predicted class' manifold exceeds a predefined threshold. As the decision is threshold-dependent, we report our results as the area under the ROC curve (AUC). We use ResNet 56 (He et al., 2016) as our classifier, and compare our results to two baselines: standard training of ResNet-56 (i.e., without any defense mechanism) and ResNet-56 equipped with the RCE defense, the current state-of-the-art model. The hyper-parameters used are listed in Appendix B.

In our study we used several attacks, which are described in Appendix E. For the bounded adversarial attack algorithms, we used two versions of FGSM and BIM, one with a small perturbation $\epsilon = 0.05$, and another with a large perturbation $\epsilon = 0.1$. For the unbounded attacks, we used JSMA and two versions of C&W: a lean version with zero confidence, and an extensive version with a higher confidence value, denoted by C&W-hc. We used the Cleverhans implementation (Papernot et al., 2018) for the attacks and applied them in an untargeted manner. A detailed description of the parameters used in the adversarial attacks appears in Appendix C. A description of the threat models we consider in this paper appears in Appendix D.

## 4.1 PERFORMANCE ON NORMAL SAMPLES

We began by evaluating the performance of our model on normal samples, as shown in Table 1. While the RCE method of Pang et al. (2018) lowered the classifier's accuracy on CIFAR-10, using the Siamese training scheme and applying the reduce variance loss increased the accuracy for both CIFAR-10 and MNIST. These results indicate that these margin-increasing procedures may be of independent value in training standard classifiers, regardless of the need for adversarial robustness.

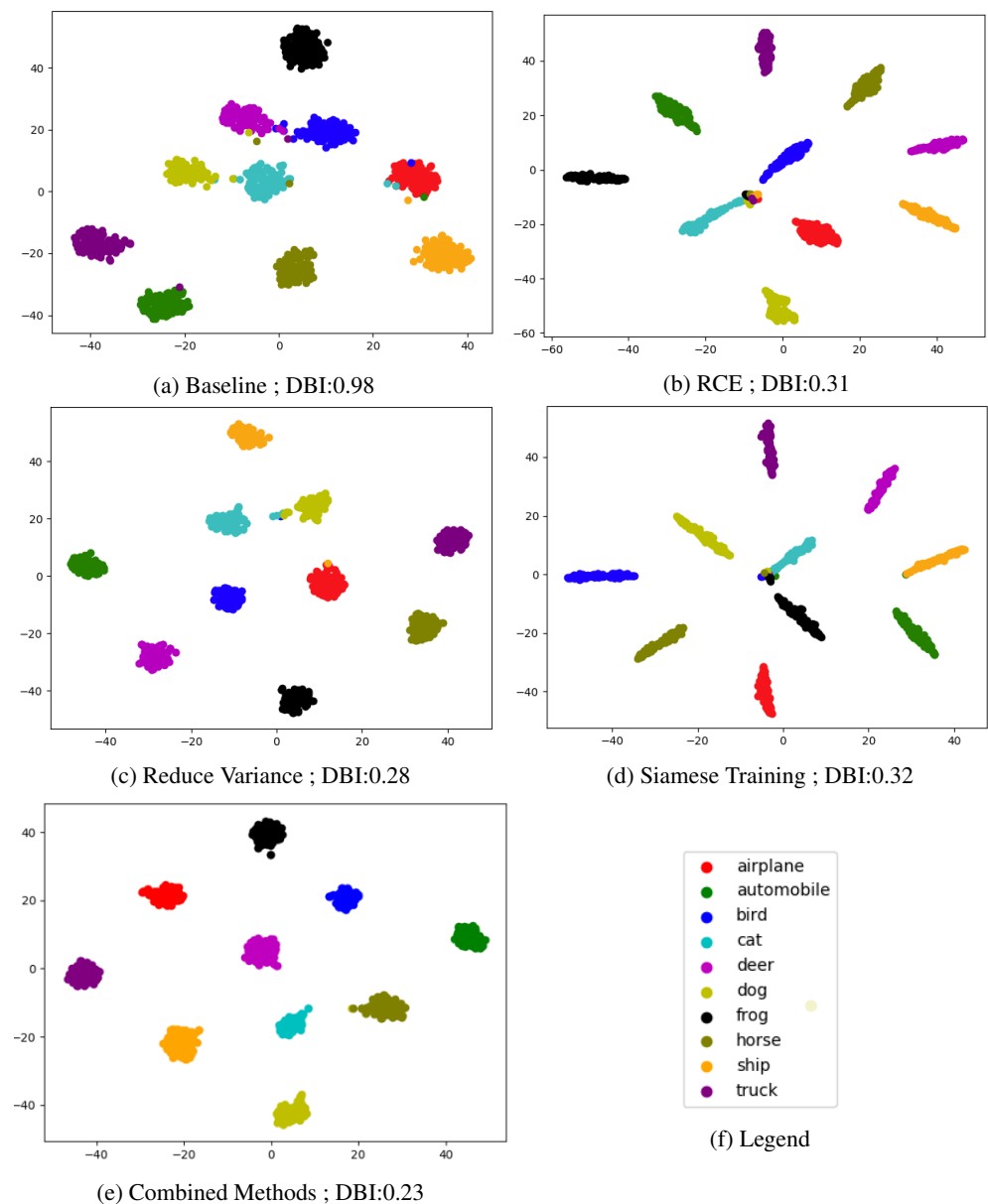

Figure 2: CIFAR-10 t-SNE visualization of the two margin increasing components of PDM . Compared to the baseline, each method contributes to the increase of the margin; the combined method display the best clustering according to the Davies–Bouldin Index.

|  | CE | RCE | Siamese + RVL | PDM (ours) |
|---|---|---|---|---|
| CIFAR-10 | 93.62 | 93.3 | 94.37 | 93.81 |
| MNIST | 99.33 | 99.32 | 99.37 | 99.52 |

Table 1: Performance on normal samples

## 4.2 GRAY-BOX MODEL

We follow a strict definition of a gray-box threat model as in (Pang et al., 2018), where the attacker has full access to the trained model, but is unaware of the detection mechanism. See Appendix D for precise definitions of the threat-models we consider in this paper. We evaluated the performance under the gray-box threat model by creating equally-sized groups of adversarial and normal examples.

We scored each example using our (KDE-based) detection mechanism. The results are shown in Table 2. The PDM detection AUC results over MNIST are outstanding, showing that all the unbounded attacks (such as C&W) were detected perfectly. On the CIFAR-10 dataset PDM performance lagged behind in defending the two BIM attacks. However, it achieved excellent AUC results for the other attacks, including a perfect score for the strong C&W high confidence attack. Figure 3 presents the resiliency of PDM and the baselines when attacked by FGSM and BIM. Consider, for example, Figure 3(a) depicting the resiliency achieved against FGSM over the CIFAR-10 dataset. The $x$-axis corresponds to the distortion step size ($\epsilon$) used by the adversary, which alters each pixel by $\pm\epsilon$. The $y$-axis measures the resiliency, namely how many perturbed instances were predicted correctly by the model. While the resiliency monotonically decreases as a function of the step size, as expected, the resiliency exhibited by PDM (blue) is consistently higher than the baselines. Similar behavior was observed for the BIM attack on this dataset. Over the MNIST dataset, PDM was more resilient than most methods, but not all step sizes.

Further investigation of the mediocre results obtained for the BIM attack revealed that the norm of the embedding layer gradients increased significantly for embedding vectors located in-between clusters. Since BIM makes a sequence of small gradient steps starting inside clusters, it is able to move further away into the center of a different class where it can no longer be detected using KDE. This observation was made by measuring the mean gradient norm after each BIM step. After several such steps we observed that the mean norm increased by an order of magnitude. This phenomenon does not occur when using solely RCE-trained model.

| | MNIST | | | CIFAR-10 | | |
|---|---|---|---|---|---|---|
| | Baseline | RCE | PDM | Baseline | RCE | PDM |
| FGSM-0.05 | 0.981 | 0.983 | **0.988** | 0.958 | 0.898 | **0.967** |
| FGSM-0.1 | 0.988 | 0.99 | **0.995** | 0.971 | 0.926 | **0.983** |
| BIM-0.05 | 0.983 | 0.967 | **0.987** | **1** | 0.99 | 0.95 |
| BIM-0.1 | 0.945 | 0.92 | **0.99** | **1** | 0.996 | 0.962 |
| C&W | 0.994 | **1** | **1** | 0.874 | 0.918 | **0.933** |
| C&W-hc | 0.88 | 0.98 | **1** | 0.637 | 0.94 | **1** |
| JSMA | 0.995 | **1** | **1** | 0.952 | 0.96 | **0.973** |

Table 2: Detection AUC under the gray-box threat model.

## 4.3 WHITE-BOX MODEL

The white-box threat model (see Appendix D) is perhaps the most interesting from the defender's viewpoint because no limitations are made regarding the information known to the attacker (Carlini et al., 2019). For the white-box threat model, we applied the C&W modified attack (hereafter referred to as C&W-wb) (Carlini & Wagner, 2017a), which has been shown to penetrate density estimation-based detection. To the best of our knowledge, this is the only known attack with this property. Following the evaluation procedure used by Pang et al. (2018), we set the parameters of the C&W-wb attack such that all attacks would succeed in fooling the model, and detection is fully breached (i.e., their AUC score $\leq 0.5$). Then, we measured the average (minimal) required distortion that was able meet these criteria. Thus, a stronger defense should yield larger distortion. We note that the distortion is quantified using the $L_2$ norm, namely, $d = \frac{1}{M} \sum_{i=1}^{M} \frac{||x_i - x_i^{adv}||_2}{\sqrt{n}}$, where $M$ is the number of adversarial instances, and $n$ is the number of pixels per image.

| | Baseline | RCE | PDM (ours) |
|---|---|---|---|
| MNIST | 0.087 | 0.104 | **0.162** |
| CIFAR10 | 0.008 | 0.019 | **0.026** |

Table 3: Distortion under the white-box threat model, scaled to [0,1], Our defense method requires 30% higher distortion on CIFAR-10 and 60% higher on MNIST

The white-box results are presented in Table 3. PDM, clearly outperforms the baselines by a wide margin by forcing a 30% higher distortion than RCE on CIFAR-10, and 60% on MNIST.

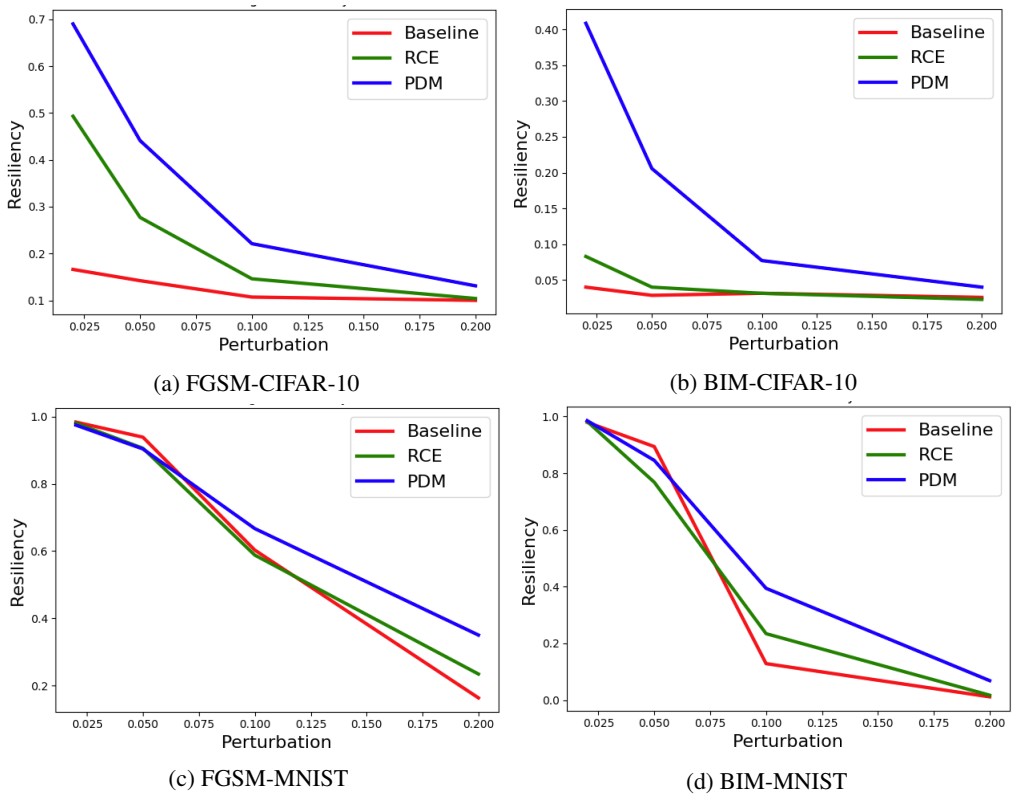

Figure 3: Resiliency under the FGSM and BIM adversarial attacks. Our method displays significantly higher resiliency

## 4.4 BLACK-BOX MODEL

To evaluate our model in the black-box setting, we follow (Papernot et al., 2017; Carlini et al., 2019) and create a proxy model, which is trained using input-output pairs probed from the defender's (target) model (i.e., the proxy model is trained via teacher-student distillation of the target model). The proxy model is then used by the attacker to generate adversarial examples under the *white-box* threat model (this black-box model variant is the most difficult, and was referred to by Pang et al. as "A white, black-box attack"). In our case, where detection is based on KDE, we could only use the C&W-wb because it is the only available attack known to penetrate KDE.

Following Pang et al. in this setting we used ResNet32 He et al. (2016) for the proxy model. Here again, we measured detection AUC rates as in the gray-box setting. The results appear in Table 4 and show that PDM is consistently and significantly better than the baselines.

|  | Baseline | RCE | PDM (ours) |
|---|---|---|---|
| MNIST | 0.88 | 0.94 | **0.99** |
| Cifar10 | 0.93 | 0.933 | **0.952** |

Table 4: Detection AUC under the black-box threat model

## 5 CONCLUDING REMARKS

We introduced a powerful approach for the defense of deep models against adversarial attacks by building on procedures for margin maximization within a penetration distortion maximization framework and the RCE loss technique. Our empirical evaluation demonstrated state-of-the-art results in defense against all threat models (with mixed results for the BIM attack). In addition, we provide some geometric intuition on attacks and defenses using t-SNE visualizations.

This work raises several interesting questions. First, it would be valuable to examine other methods for margin maximization and Jacobian reduction. For example, recently Elsayed et al. (2018) proposed a sophisticated loss function that tends to maximize the embedding margin. Similarly, a recent work by Zhang et al. (2019) proposed an iterative technique to reduce the norm of the Jacobian. Finally, it would be very interesting to explore ways to increase the margin (and reduce the Jacobian) on shallower embedding layers where lower-level features are formed.

**Acknowledgment:** The authors would like to express their gratitude to Tianyu Pang for his invaluable help in the implementation and understanding of his work on the reverse cross-entropy method.

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

Appendix

## A  $L_2$ NORM SUB-MULTIPLICATIVITY

Given $A \in \mathbb{R}^{NxMxK}$ and $B \in \mathbb{R}^{KxJxL}$, we claim that the Frobenius norm of the multiplication of $A$ and $B$ is less than or equal to the multiplication of each tensor's Frobenius norm. Proof:

$$\|AB\|_F^2 = \sum_{n=1}^{N} \sum_{m=1}^{M} \sum_{j=1}^{J} \sum_{l=1}^{L} \left| \sum_{k=1}^{K} a_{n,m,k} b_{k,j,l} \right|^2$$

$$\leqslant \sum_{n=1}^{N} \sum_{m=1}^{M} \sum_{j=1}^{J} \sum_{l=1}^{L} \sum_{k=1}^{K} |a_{n,m,k}|^2 \sum_{k=1}^{K} |b_{k,j,l}|^2 \qquad \text{(Cauchy-Schwarz)}$$

$$= \sum_{n=1}^{N} \sum_{m=1}^{M} \sum_{j=1}^{J} \sum_{l=1}^{L} \left( \sum_{k,s=1}^{K} |a_{n,m,k}|^2 |b_{k,j,l}|^2 \right)$$

$$= \sum_{n=1}^{N} \sum_{m=1}^{M} \sum_{k=1}^{K} |a_{n,m,k}|^2 \sum_{j=1}^{J} \sum_{l=1}^{L} \sum_{s=1}^{K} |b_{s,j,l}|^2 \qquad (6)$$

$$= \|A\|_F^2 \|B\|_F^2$$

## B  CLASSIFIER HYPER-PARAMETERS

| Parameter | Value |
|---|---|
| Optimizer | SGD |
| ResNet Depth | 56 |
| Weight Regularization | L2 (0.002) |
| Batch Size | 128 |
| Initial Learning Rate | 0.1 |
| Epochs-CIFAR-10 | 400 |
| Epochs-MNIST | 80 |
| Activation | Leaky-Relu (0.1) |

## C  ADVERSARIAL ATTACK PARAMETERS

| Attack | Parameter | Value |
|---|---|---|
| BIM | Iterations | 10 |
| JSMA | Max Iterations | 100 |
| CW | Max Iterations | 10 |
|  | Binary Search Steps | 9 |
|  | Confidence | 0 |
| CW-hc | Max Iterations | 1000 |
|  | Binary Search Steps | 9 |
|  | Confidence | 10 |
| CW-wb | Max Iterations | 5000/10000 |
|  | Confidence | 0/0 |
|  | Binary Search Steps | 3/3 |

Table 5: Adversarial attacks parameters

## D    THREAT MODELS

Rigorous analyses of adversarial defense systems requires precise specification of what the adversary knows and can do. Conducting this type of specification is called *threat modeling* (Kurakin et al., 2016). In this paper we consider the three core models, which are differentiated by their knowledge of the classifier (target network) and the defense mechanism being used.

- **Black-box**: In this weakest scenario, the adversary has no knowledge of our system. It does not know the target network architecture, cannot access its gradients and does not know the defense methods. The adversary can, however, sample the targeted network for input-output pairs and has access to the dataset used to train the target network.

- **Gray-box** (a.k.a. oblivious): Here the adversary has full knowledge of the target network and can access its gradients and parameters, including the data being used in the training process. However it has no information on the defense mechanism.

- **White-box**: In this most challenging scenario, we assume the adversary has full knowledge of the entire system to be attacked, which includes the target network, the defense method, their parameters and the data used for training.

To ensure a clear distinction between threat models, Carlini et al. (2019) recently proposed a comprehensive methodology for evaluating adversarial attacks and defenses, thus emphasizing the difference between gray-box and white-box attacks. We believe that following the Carlini et al. guideline is crucial for advancing this line of research and therefore we adhere to them strictly.

## E    ATTACK ALGORITHMS

Adversarial attack algorithms aim to fool DNNs such that a given image is misclassified with minimal perturbations. In *untargeted attacks*, the adversary aims to minimize the true class activation so that another arbitrary class will be predicted. In this paper, we focus on *targeted attacks* where the adversary aims to fool the classification in a controlled manner such that a specific class will be predicted instead of the correct one. Thus, targeted attacks are considered far more dangerous. For example, fooling an autonomous driving system to interpret a *stop* sign as a *speed limit* sign is far more dangerous than interpreting it as a *yield* sign.

In this section, we discuss the various attack algorithms we use to evaluate our defense method. We denote $x, x'$ as the input and adversarial images, respectively, $\ell$ as the target label, $F$ as the target model with loss function $L_F(x, \ell)$ and $\epsilon = ||x - x'||$ as the pixel-wise perturbation between the adversarial and normal images. The general formulation, therefore, becomes,

$$\underset{x'}{\text{minimize}} \ ||x - x'||^2 \ \ s.t. \ F(x') = \ell \tag{7}$$

**FGSM**

Goodfellow et al. (2014) introduced the fast gradient sign method (FGSM), which optimizes the adversarial image by back-propagating the input through the attacked DNN, in accordance with the desired target. Formally, letting $\epsilon$ be a fixed parameter, the adversarial example is,

$$x' = x + \epsilon \, \text{sign}(\nabla L_F(x, \ell) \tag{8}$$

While not as effective as other attack algorithms, this method has the advantage of being one of the fastest ones.

**BIM**

Kurakin et al. (2016) introduced the Basic Iterative Method (BIM), which performs the FGSM method iteratively, clipping the perturbation if needed. Formally,

$$x'_{N+1} = x'_N + \epsilon \, \text{sign}(\nabla L_F(x'_N, l)),$$

where $\epsilon$ is a fixed parameter.

**C&W**

Carlini & Wagner (2017c) introduced the C&W method, which operates by modifying the L-BFGS method as follows,

$$\underset{x'}{\text{minimize}} \ ||x - x'||^2 + cf(x'),$$

where $c$ is a hyperparameter and the loss function, $f$, is chosen such that $f(x') <= 0$ if $x'$ is classified as the target class; namely,

$$f(x') = \max(Z(x')_\ell - Z(x')_t, \kappa)$$

where $Z$ is the target DNN logits, $t$ is the correct label and $\kappa$ is a hyperparameter referring to the confidence of the attack. The higher the confidence, the higher the activation for the target class is, and therefore, the larger the perturbation.

Three different Euclidean norms are considered with this algorithm, $L_0, L_2, L_\infty$. Following Pang et al. (2018), we conduct our evaluation using $L_2$. This attack method is considered very effective and has had great success in overcoming various defense methods Carlini & Wagner (2017c),Papernot et al. (2016b).

**JSMA**

Papernot et al. (2016a) introduced the Jacobian-based Saliency Map Attack (JSMA) which alters a single pixel of $x$ at each iteration to maximize the saliency map. This method is known to enforce large perturbations but on fewer pixels than other methods.

## F  PDM TRAINING ALGORITHM

---
**Algorithm 1** PDM Training
---
1: **procedure** PDM
2:     **for** batch $= 1, \ldots,$#batches **do**
3:         $X, Y \leftarrow$ get_batch()
4:         initialize $X_{\text{siamese}} = [], Y_{\text{siamese}} = [], S = []$
5:         **for** $b = 1, \ldots,$ batch_size **do**
6:             $q \sim$ Bernoulli$(Q)$
7:             **if** $q == 1$ **then**
8:                 $y \leftarrow Y[b]$
9:                 $s \leftarrow 1$
10:             **else**
11:                 $y \leftarrow$ random class $\neq y_1$
12:                 $s \leftarrow 0$
13:             $x \leftarrow$ random sample from class $Y[b]$
14:             Append $x, y, s$ to $X_{\text{siamese}}, Y_{\text{siamese}}, S$
15:         $z_1, z_2 \leftarrow F_\ell(X), F_\ell X_{\text{siamese}})$          $\triangleright$ sample's embedding
16:         $p_1, p_2 \leftarrow F(X), F(X_{\text{siamese}})$
17:         SL $= \frac{1}{\text{batch\_size}} \sum |\frac{z_1 z_2}{||z_1||||z_2||} - S|$          $\triangleright$ model logits
18:         $RVL \leftarrow 0$
19:         **for** $c = 0, \ldots,$classes **do**
20:             $\mu_c = \frac{1}{N_c} \sum_{i=1}^{N_c} z_i^c$
21:             $\sigma_c = \frac{1}{N_c} \sum_{i=1}^{N_c} ||z_i^c - \mu_c||_2$
22:             RVL = RVL + $\sigma_c$
23:         minimize $-R^Y \log(p_1) - R^{Y_{\text{siamese}}} \log(p_2) +$ RVL + SL
---

