# OpenReview forum: "Improved Detection of Adversarial Attacks via Penetration Distortion Maximization"
_ICLR.cc/2020/Conference — Reject_

### Official Review · AnonReviewer3 · 2019-10-15
**Official Blind Review #3**

**Rating:** 3

**Review:**

After rebuttal: my rating remains the same.
I have read other reviewers' comments and the response. Overall, the contribution of retraining and detection with previously explored kernel density is limited.

=================
Summary:
This paper proposes new regularization techniques to train DNNs, which after training, make the crafted adversarial examples more detectable. The general idea is to minimize the inter-class variance and maximize the intra-class distance, at some feature layer. This involves regularization terms: 1) SiameseLoss, an existing idea of contrastive learning known can increase inter-class margin; 2) reduce variance loss (RVL), a variance term on deep features, and 3) reverse cross entropy (RCE), a previously proposed term for detection purpose. The motivation behind seems intuitive and the empirical results demonstrate moderate improve in detection AUC, compared to one existing technique (e.g RCE).

My concerns:
1. The proposed technique requires retraining the networks to get a few percents of detection improvement. This is a disadvantage compared to standard detection approaches such as [1] and [2] which do not need to retain the network. I am surprised that these standard detection methods were not even mentioned at all. Retraining with fixed loss becomes problematic when the networks have to be trained using their own loss functions due to application-specific reasons. Moreover, the detection performance reported in this paper is not better than the one reported in [2] (ResNet, CIFAR-10, 95.84%) which do not need retraining.

2. There are already well-known margin-based loss functions, such as triplet loss [4], center loss [5], large-Margin softmax loss [6], and many others, which are not mentioned at all.

3. In terms of retraining-based detection, higher AUCs have been reported in [3] for a neural fingerprinting method.

4. Incorrect references to existing works. The second sentence in Intro paragraph 2: Metzen, et al, .... these are not adversarial training. Xu, et al. (feature squeezing) is not a randomization technique.

5. The "baseline" method reported in Table 2, is confusing. RCE is also a baseline? You mean conventional cross entropy (CE) training?

6. Some of the norms are not properly defined, which can be confusing in adversarial research. For example, from Equation (1) to (4). The "Frobenius norm used here" statement in Equation (3), don't know this F norm comes from.


[1] Characterizing adversarial subspaces using local intrinsic dimensionality. ICLR, 2018
[2] A simple unified framework for detecting out-of-distribution samples and adversarial attacks. NeurIPS, 2018
[3] Detecting Adversarial Examples via Neural Fingerprinting. arXiv preprint arXiv:1803.03870, 2018
[4] Facenet: A unified embedding for face recognition and clustering. CVPR, 2015.
[5] A Discriminative Feature Learning Approach for Deep Face Recognition. ECCV, 2016.
[6] Large-Margin Softmax Loss for Convolutional Neural Networks. ICML 2016.

**Experience Assessment:**

I have published one or two papers in this area.

**Review Assessment: Checking Correctness Of Derivations And Theory:**

I carefully checked the derivations and theory.

**Review Assessment: Checking Correctness Of Experiments:**

I carefully checked the experiments.

**Review Assessment: Thoroughness In Paper Reading:**

I read the paper thoroughly.

---

> ### Author Response · Authors · 2019-11-12
> **Reply to  Reviewer #3**
>
> Thanks for your thoughtful comments. Please consider our response.
>
> 1. "The proposed technique requires retraining."
>
> Our method intends to be attack-agnostic and is thus compared only to such methods.
> We agree that it would be most interesting to remain attack-agnostic without retraining.
> The two papers you mentioned ([1] and [2]) clearly utilize adversarial optimization (i.e., using adversarial examples for optimization) and are thus not attack-agnostic. Therefore, this is not a fair comparison. To verify that [1] and [2] utilize adversarial optimization,see section 5.2 in [1] and section B2 in supplementary material in [2]
>
> 2. "There are already well-known margin-based loss functions, such as triplet loss [4], center loss [5], large-Margin softmax loss [6], and many others, which are not mentioned at all."
>
> Thanks for the references. We plan to experiment with some of them in future work. We will mention all these margin increasing loss functions.
>
> 3. In terms of retraining-based detection, higher AUCs have been reported in [3] for a neural fingerprinting method.
>
>
> We agree that the Neural Fingerprinting paper is very interesting and presents phenomenal results. However, the threat model of that paper is completely different than in our setting. Specifically, in their gray-box threat model, the adversary has no information about the fingerprint instances whatsoever, which amounts to several thousands secret parameters. In our gray-box threat model the adversary is solely unaware of the use of the KDE-based defense mechanism. Thus, no apples-to-apples comparison can be made here.
>
> In the white-box threat model, we measure the *distortion* required to fool our model using the CW-wb attack, while the fingerprint paper they implemented an adaptive version of FGSM, BIM and SPSA and measured *robustness* to a given set of hyper-parameters. Here again, the comparison between the two papers isn't apples-to-apples.
>
> We note that our threat models follow the ones defined in [1][2]
> [1]Feinman, et,al. Detecting adversarial samples from artifacts
> [2]Pang et,al. Towards robust detection of adversarial examples
>
> 4.  "Incorrect references:
> Fixed
>
> 5. "RCE is also a baseline?"
> Yes. To the best of our knowledge the RCE NIPS-2018 paper still presents SOTA results for adversarial detection.
>
> 6. "Some of the norms are not properly defined"
>
> While we could use any Lp norm, all our results are achieved with the L2 norm for the embedding  and the Frobenius for the Jacobian. Fixed.

---

### Official Review · AnonReviewer2 · 2019-10-16
**Official Blind Review #2**

**Rating:** 3

**Review:**


Summary
========
This paper proposes a defense against adversarial examples that detects perturbed inputs using kernel density estimation. The paper uses a combination of known (and often known to be broken) techniques, and does not provide a fully convincing evaluation.
I lean towards rejection of this paper.

Detailed comments
=================
The idea of increasing robustness by maximizing inter-class margins and minimizing intra-class variance is fairly natural, but the author's discussion of their approach (mainly in sections 1 and 2) is very hand-wavy and relies on a lot of general intuitions and unproven claims about neural networks.

For example, in the introduction, the authors claim:

"A trained deep classification model tends to organize instances into clusters in the embedding space, according to class labels. Classes with clusters in close proximity to one another, provide excellent opportunities for attackers to fool the model. This geometry explains the tendency of untargeted attacks to alter the label of a given image to a class adjacent in the embedding space as demonstrated in Figure 1a."

First, a t-SNE representation is just a 2D projection of high-dimensional data that is useful for visualization purposes, and one should be careful when extrapolating insights about the actual data from it. For example, distances in the 2D projection do not necessarily correspond directly to distances in the embedding space.
The claim that untargeted attacks lead to a "nearby" cluster are hard to verify given just Figure 1. First, the colors of the labels between 1a and 1b do not seem to match (e.g., Dog is bright green in 1b but this color does not appear in 1a). If the other colors match, then this would seem to suggest that trucks (purple) often get altered to ships (orange). Yet, the two clusters are quite far apart in 1a. It seems hard to say something qualitative here. An actual experiment comparing distances in the embedding space and the tendency of untargeted attacks to move from one class to another would be helpful.
The color scheme in Figure 1b is also unclear. A color bar would help here at the very least.

These observations are then used to justify increasing cluster distance while minimizing cluster variance, but it would be nice to see a more formal argument relating these concepts to the embedding distance.

The technique proposed in Section 3.2. to reduce variance loss estimates each class' variance on each batch. Would  this still work for a dataset with a large number of classes (e.g., ImageNet)? For such a dataset, each class will be present less than once in expectation in each batch, which seems problematic.

The plots in Figure 2 don't give much of a sense of how the combination of the different proposed techniques is better than any individual technique. The evaluation compares PDM to RCE, but from Figure 2 one could guess that variance reduction alone (2c) performs very similarly to PDM (2e). An ablation study showing the contribution of each of the individual techniques would be helpful.

The evaluation section could be improved significantly. FGSM, JSMA, and to some extent BIM, are not recommended attacks for evaluating robustness. The gray-box and black-box threat model evaluations are also not the most interesting here. Instead, and following the recommendations of Carlini et al. (2019), the evaluation should:

- Propose an adaptive attack objective, tailored for the proposed defense in a white-box setting. The authors do this to some extent, by re-using the attack objective from Carlini & Wagner 2017, which targets KDE. It would still be good to provide additional explanations about how the hyperparameters for this attack were set.
- Optimize this objective using both gradient-based and gradient-free attacks
- As the proposed defense is attack-agnostic, I also suggest trying it out on rotation-translation attacks, as the worst-case attack can always be found by brute-force search

Other
=====
- The citations for adversarial training in the 2nd paragraph of the intro are unusual. Standard references here are for sure the first two below, and maybe some of the other three as is relevant to your work
    - Szegedy et al. 2013: "intriguing properties of neural networks"
    - Goodfellow et al. 2014: "Explaining and harnessing adversarial examples"
    - Kurakin et al. 2016: "Adversarial Machine Learning at Scale"
    - Madry et al. 2017: "Towards deep learning models resistant to adversarial attacks"
    - Tramer et al. 2017: "Ensemble Adversarial Training"
- The Taylor approximation in (1) does not seem to be well defined. The Jacobian of F is a matrix, so it isn't clear what evaluating that matrix at a point x means.
- The "greater yet similar" symbol (e.g., in equation (4)) should be defined formally.

**Experience Assessment:**

I have published in this field for several years.

**Review Assessment: Checking Correctness Of Derivations And Theory:**

I carefully checked the derivations and theory.

**Review Assessment: Checking Correctness Of Experiments:**

I carefully checked the experiments.

**Review Assessment: Thoroughness In Paper Reading:**

I read the paper thoroughly.

---

> ### Author Response · Authors · 2019-11-12
> **Reply to  Reviewer #2**
>
> Thanks for your thoughtful comments. Please consider our response.
>
> 1.  "relies on a lot of general intuitions and unproven claims about neural networks." "hard to verify given just Figure 1."
>
> There was indeed a problem with the color scheme - now fixed (in the new version).
> We followed your advice and conducted an analysis showing that these claims directly hold on the embedding space (and not relying on t-sne). The conclusion is qualitatively the same. When considering L2 distance in the embedding space 70% of the attacks on instance x targets one of the two closest classes to x. This will be added to the next version.
>
> 2. Variance reduction: "Would  this still work for a dataset with a large number of classes (e.g., ImageNet)?"
>
> We have checked the variance reduction on Cifar-100 (100 classes), and found that it still works.
> Specifically, when examining the embedding clustering quality using the Davies-Bouldin index (DBI) we get an improvement of of 25%
> These preliminary results will be included after the rebuttal.
>
> 3. Evaluation section
>
> (a) "It would still be good to provide additional explanations (white-box model)"
>
> We followed the procedure for the KDE spoofing attack in which one sets the hyper-parameters such that all generated adversarial examples are able to fool the targeted model. The required such hyper-parameters are specified in Appendix C.
>
> (b) "Optimize this objective using gradient-free attacks"
>
> We included the SPSA gradient-free attack on Cifar-10. The results indicate better performance using our method (significantly better in the case of resiliency). For a perturbation (epsilon) of 0.05, PDM achieved a robustness (adversary fail rate) of 0.4 compared to 0.13 achieved by an RCE trained model and 0.08 achieved by a CE trained model.
>
> (c)  "I also suggest trying it out on rotation-translation attacks"
>
> Done. And results are consistent. Our method is still much better.
> Specifically, PDM achieved an AUC score of 0.931 compared to 0.914 achieved by an RCE trained model and 0.89 achieved by a CE trained model.
>
> 4. "Missing citations."
> Thanks, added. The new version now includes them all.

---

### Official Review · AnonReviewer1 · 2019-10-24
**Official Blind Review #1**

**Rating:** 6

**Review:**

Update after author response:
I would like to thank the authors for the thoughtful response, and for addressing some of the concerns raised by the reviewers. The draft appears improved but my concerns about the novelty and interpretability of the work still stand, leading me to keep my assessment unchanged.
---------------------------

In this paper, the authors propose a general defense method against adversarial attacks by maximizing an approximate bound on the magnitude of distortion needed to force a misclassification. The authors note that this maximization can be achieved by increasing the margin between class clusters and by reducing the norm of the Jacobian of intermediate layers. Subsequently, they either directly adopt or introduce simple modifications to existing techniques to affect these two factors, showing the robustness of the combined method to several adversarial attacks on MNIST and CIFAR-10 datasets.

As neural networks get deployed for increasingly critical applications, the issue of defense against adversarial attacks becomes progressively relevant. The paper does a good job of motivating a relatively simple approach to the problem based on an approximate bound, and pulls in from different existing methods to build a robust system. The strong points of the paper:
1. The paper is clearly written, and the approach is sensible.
2. Fairly thorough empirical investigation under different threat models.
3. The proposed method performs consistently above the baselines for different experiments.

Here are some of my concerns:
1. The work is somewhat incremental and the novelty mostly lies in pulling a few different methods together that seem to work well in unison.
2. The two methods used for increasing the margin don’t actually optimize that objective directly. The Siamese Loss uses cosine distance as proxy and the variance reduction doesn’t guarantee increase in margin which is sensitive to outliers. Any improvement achieved thus appears to be an ill-understood side-effect.
3. The definition of cluster distance (page 3) looks erroneous.
4. The authors note that the proposal doesn’t work very well for a specific kind of attack (BIM) but don’t have clear recommendations for improvement. The tentative explanation of why this happens is also somewhat loose.

In summary, I think the paper addresses an interesting problem even though the development is arguably incremental. However, since the unified approach is simple yet novel, and the results fairly promising, I am somewhat inclined to accept this paper.

**Experience Assessment:**

I do not know much about this area.

**Review Assessment: Checking Correctness Of Derivations And Theory:**

I assessed the sensibility of the derivations and theory.

**Review Assessment: Checking Correctness Of Experiments:**

I carefully checked the experiments.

**Review Assessment: Thoroughness In Paper Reading:**

I read the paper thoroughly.

---

> ### Author Response · Authors · 2019-11-12
> **Reply to Review #1**
>
> Thanks for your thoughtful comments. Please consider our response.
>
> 1. We agree that the novelty of our method is in the combination of several techniques that work well in unison. We note that the combination itself is intuitive and, more importantly, it leads to significant and consistent improvements.
>
> 2. While it would be interesting to explore other metrics,
> the use of cosine as a proxy is common many applications. Our choice followed the need to use a differentiable and bounded metric.
>
> In regards to variance reduction, our intention (which will be clarified in the next version) is to consider the class-wise *average* variance, which is less sensitive to outliers. When considering the average the proposed method works very well.
>
> 3. Cluster distance definition is indeed wrong - now fixed (in the new revision).
>
> 4. BIM deficiency: We agree it is a deficiency. We expect it however to disappear when using a different Jacobian smoothing method. On the positive side, we have made a significant step toward identifying the source of this problem (see preliminary explanation in Section 4.2),
> which we plan to resolve.

---

### Decision · Program_Chairs · 2019-12-19

**Decision:**

Reject

**Comment:**

A defense against of adversarial attacks is presented, which builds mostly on combining known methods in a novel way. While the novelty is somewhat limited, this would be fine if the results were unequivocally good and other parts of the problematic. However, reviewers were not entirely convinced by the results, and had a number of minor complaints with various parts of the paper.

In sum, this paper is not currently at a stage where it can be accepted.